# MODELLING OPTIMAL TRADE-OFF BETWEEN CONTINUED PRE-TRAINING AND SUPERVISED FINE-TUNING FOR LLM DOMAIN ADAPTATION

## ABSTRACT

Domain adaptation is critical for tailoring pre-trained Large Language Models (LLMs) to specialised tasks without significant costs of pre-training from scratch. Two common approaches for domain adaptation are Continual Pre-training (CPT) and Supervised Fine-Tuning (SFT), yet the data mix for each is often determined arbitrarily based on data availability or through limited data ablations. In this paper, we present a mathematical framework to model downstream domain performance as a function of the ratio between CPT and SFT under a fixed token budget. Using 7B-parameter pre-trained LLMs, we perform domain adaptation training across three domains - health, chemistry, and coding - within a 30B-token limit. CPT uses domain-relevant subsets of Nvidia's ClimbLab dataset, while SFT employs medqa (health), OpenCodeInstruct (programming), and ChemData700k (chemistry). Resultant models are evaluated on domain-specific QA benchmarks across sixteen CPT:SFT allocations. Results show that optimal performance, regardless of domain, arises from allocations with effective CPT:SFT token ratios between 29.9976B:2.4M and 29.9982B:1.8M corresponding to a CPT fraction of approximately 0.99992 - 0.99994. Our optimal split demonstrated an **11.6 % score improvement** over the state-of-the-art domain-adapted model Code Llama and a **6.4 % increase in performance** on MedQA over HippoCrates Meta 7B while approaching the performance of HippoCrates Mistral 7B, at up to **95% token budget reduction.** We further validate these findings through ablation with trained models to better understand the impact of individual datasets on resultant model weights. Our work provides a framework for guiding efficient domain adaptation of LLMs through CPT and SFT.

## 1 INTRODUCTION

Large Language Models (LLMs) have consistently achieved noteworthy performance across a wide range of Natural Language Processing (NLP) tasks. Despite being trained on, their downstream performance on domain-specific tasks often requires specialised adaptation, particularly in well-documented domains of Chemistry, Medicine, and Coding. Domain-adapted models are widely used in legal, healthcare, and government affairs domains as knowledge query chatbots or to generate domain-specific texts (Downie & O'Brien, 2024; Nazi & Peng, 2024). To bridge the gap and achieve state-of-the-art performance for domain adaptation, LLMs often undergo two complementary stages: Continual Pre-Training (CPT) and Supervised Fine-Tuning (SFT).

CPT pre-trains a model on unlabeled domain-specific corpora for knowledge transfer (Ke et al., 2023; Gururangan et al., 2020), while SFT fine-tunes the model on task-specific, labelled data to get the desired outputs from the LLMs (Ouyang et al., 2022). Both CPT and SFT, with their respective training data, have been shown to significantly improve downstream performance for domain-specific benchmarks (Sun & Dredze, 2024). Many open-source, domain-adapted LLMs are available, and training recipes for the CPT:SFT split have been documented in domains such as Chemistry, Medicine, and Coding (Thirunavukarasu et al., 2023; Yu et al., 2024). Examples include Hippocrates-Mistral 7B, which leveraged 298M medical tokens for CPT and 58.7M instruction tokens for SFT (De et al., 2024), while CodeLlama used up to 600B tokens during CPT (Rozière et al.,

2023). These models typically use curated training data without constraints to maximise downstream performance.

Despite these advancements, high-quality SFT datasets are scarce due to the cost and time required to create them(Liu et al., 2024a; Wang et al., 2025a). Consequently, most studies aggregate all available SFT data to maximise downstream performance and arbitrarily allocate CPT:SFT based on the volume of data collected, often using all available sources for each phase, without considering redundancy or the optimal ratio for downstream performance.

However, this raises the question: how should limited domain-specific data be allocated between CPT and SFT to maximise downstream performance? Previous research has investigated the optimal mixture of general and domain-specific data for CPT in LLMs, with the aim of balancing efficiency and training costs (Que et al., 2024). To the best of our knowledge, this is the first study of the trade-offs between CPT and SFT. Most current approaches rely on arbitrary splits.

In this paper, we address this gap by exploring the relationship and finding an optimal ratio between the CPT:SFT ratio split constrained under a fixed total token budget, bounded by the maximum SFT dataset size. We performed domain-adaption training for a 7B-parameter pre-trained LLM with 30B token budget across Chemistry, Medicine, and Coding, varying CPT:SFT token allocation and evaluating downstream performance on their respective benchmarks. Key contributions of this study include: (1) showing that downstream performance across all three domains consistently peaks around the same narrow CPT:SFT range (2) identifying the optimal ratio is in the range of 29.9976B:2.4M and 29.9982B:1.8M tokens(3) demostrating empirical data can be fitted with a hybrid mathematical model and validated through experiments across varying total token budgets from 25B to 35B.

## 2 RELATED WORK

Prior work has investigated the individual impacts of Continual Pre-training (CPT) and Supervised Fine-Tuning (SFT), establishing them as complementary pillars of domain adaptation. CPT is highly effective for imbuing generalist models with specialised knowledge from large, unlabeled corpora Gururangan et al. (2020). Work by Gururangan et al. demonstrated that CPT serves as a critical preparatory step, significantly enhancing the efficacy of subsequent task-specific tuning, particularly in scenarios with domain shifts. This established a foundational understanding that CPT prepares a model for SFT by aligning its representations with the target domain.

Building on this, recent research has focused on optimising the CPT phase itself. recognising the high costs associated with extensive pre-training, Que et al. (2024) proposed a "D-CPT Law," a scaling law designed to predict model performance based on the mixture of general versus domain-specific data. This work aims to find an optimal balance within CPT itself to maximise domain knowledge acquisition while mitigating the risk of "catastrophic forgetting" of the model's general capabilities, thereby avoiding expensive trial-and-error ablations.

The roles of CPT and SFT were further analysed by Sun et al. (2024), who found that SFT provides minimal benefit for tasks already well-represented in the pre-training data Sun & Dredze (2024). However, for tasks outside the pre-training distribution, SFT delivered substantial gains. Their findings suggest a point of diminishing returns for CPT, after which SFT becomes a more efficient mechanism for performance improvement. This highlights a critical trade-off: CPT is essential for building a knowledge base, while SFT is crucial for directing that knowledge toward specific tasks.

Despite this progress, the strategic allocation of a fixed data budget between these two phases remains a significant gap in the literature. Most studies either assume unlimited token budget or focused on optimising the data mixture within CPT. Consequently, the precise CPT:SFT ratio is often determined by practical constraints like SFT data availability rather than a principled strategy Liu et al. (2024a). The literature lacks a systematic, empirical analysis of how downstream performance is affected when every token allocated to CPT is one less token available for SFT, and vice versa. This study directly addresses this gap by investigating the CPT:SFT trade-off under a fixed token budget, seeking to find the optimal ratio and mathematical model.

## 3 METHODOLOGY

### 3.1 EXPERIMENTAL DESIGN

Mistral 7Bv0.3 (Jiang et al., 2023) was used for CPT and SFT phases. Both stages were implemented using the open-source Mistral fine-tune package (Mistral, 2024).

#### 3.1.1 CONTINUAL PRE-TRAINING

For domain adaptation in specialised domains, a Parameter-Efficient Fine-Tuning (PEFT) strategy with Low-Rank Adaptation (LoRA) was employed for the CPT stage. The training recipe for CPT is summarised in Appendix Table A.2. While prior studies often use large step counts (Wang et al., 2025b; Que et al., 2024), with Appendix Table A.1 ablations with 4K–10K steps showed 2K steps sufficed, reducing compute while maintaining performance. Nvidia's Climb dataset (Diao et al., 2025) clusters 7, 11, and 17 were used ($< 30B$ tokens per domain; Table A.4).

#### 3.1.2 SUPERVISED FINE-TUNING

Following the CPT stage, supervised fine-tuning (SFT) was conducted using instruction-based datasets specific to each domain, with the primary objective of aligning the model's domain knowledge to the desired task format. The training recipe for SFT is summarised in Appendix Table A.3. Referring to Table A.5, training data originated from domain-specific instruction-tuning datasets (BigBio MedQA (Jin et al., 2021) for the health domain, Open Code Instruct for Python coding (Ahmad et al., 2025), and ChemData700k for chemistry (Zhang et al., 2024).

#### 3.1.3 BENCHMARK DATASETS

We evaluate our trained models with four domain-specific benchmarks: HumanEval, HumanEval Plus, ChemBench4k, MedQA, and on two general-purpose benchmarks: MMLU and HellaSwag (see Appendix B for full details).

### 3.2 DATASET SPLITTING STRATEGY

Most LLM domain adaptation data selections for CPT or SFT are often made arbitrarily or greedily. Researchers typically aggregate all available domain-specific datasets, sometimes combining multiple sources without considering redundancy or the relative importance of each subset. It is unclear whether performance is driven by the sheer volume of CPT/SFT data or by the precise allocation of the overall budget. Without clarity, dataset sizes risk being unnecessarily inflated, leading to wasted compute during training.

Our work challenges the "more is better" assumption for SFT data. We operate under a realistic, resource-constrained regime where the high-quality SFT dataset is intentionally small, capped at a maximum of **2.4 million tokens**. This allows us to directly study the trade-off between building foundational knowledge via CPT and SFT. Rather than an arbitrary split, we use an empirically-driven approach to find the optimal balance. Pairing a large CPT corpus with a small, purposeful SFT dataset proves highly efficient, achieving competitive results.

### 3.3 MATHEMATICAL MODELING FRAMEWORK

To provide a theoretical explanation for our empirical findings, we developed a mathematical framework to model the relationship between the SFT data budget and downstream performance.

We propose a hybrid model that combines two distinct mathematical components: a Gaussian function to model the performance peak and a reciprocal function to model the alignment cliff. The model is formulated as follows:

$$\hat{P}(S) \;=\; P_{\text{base}} + A \cdot \exp\left(-\tfrac{1}{2}\left(\tfrac{\ln S - \mu}{\sigma}\right)^2\right) - \frac{\lambda}{S - S_{\min}}, \tag{1}$$

where $\hat{P}(S)$ denotes the predicted downstream performance as a function of the number of SFT tokens $S$, and:

- $S$ = number of SFT tokens,
- $P_{\text{base}}$ = baseline performance,
- $A$ = amplitude of the Gaussian peak,
- $\mu$ = location of the peak in log-SFT space,
- $\sigma = e^{\log \sigma}$ = width of the Gaussian,
- $S_{\min}$ = lower bound for SFT tokens (cliff threshold),
- $\lambda$ = penalty strength.

where $\hat{P}(S)$ is the predicted performance for a given number of SFT tokens, $S$.

The first term in Eq. 1 represents the **Gaussian Peak**, which models the primary performance curve as a Gaussian in the logarithmic space of SFT tokens. This captures the rise to an optimal point and the subsequent diminishing returns, with $\mu$ denoting the centre of the peak in log-SFT space, corresponding to the optimal number of SFT tokens. The second term is the **Reciprocal Cliff**, which models the sharp performance collapse when the number of SFT tokens falls below a critical threshold $S_{\min}$. This penalty term diverges as $S$ approaches $S_{\min}$, reflecting an "alignment failure."

The free parameters of the model—including the baseline performance ($P_{\text{base}}$), peak amplitude ($A$), peak location ($\mu$), peak width ($\sigma$), cliff location ($S_{\min}$), and penalty strength ($\lambda$)—were calibrated by fitting Equation 1 to our 30B budget experimental data using a non-linear least squares Optimiser.

**Predictive Application.** The core hypothesis of this framework is that the optimal number of SFT tokens, $N_{\text{SFT, opt}} = \exp(\mu)$, is a constant property of the adaptation task. Once this value is determined through calibration, the model can be used to predict the optimal CPT fraction for any given total token budget, $N_{\text{total}}$, using the simple arithmetic relationship:

$$\text{CPT Fraction}_{\text{opt}} = \frac{N_{\text{total}} - N_{\text{SFT, opt}}}{N_{\text{total}}} \tag{2}$$

We conducted further experiments to demonstrate the scalability of this model by varying the total token budget.

## 4 RESULTS

Through experiments on downstream domain-specific benchmarks (Table 6 and Figure 1), we demonstrate the effectiveness of our optimised data split across the Chemistry, Health, and Coding domains. As shown in Figures 1a, 1c, and 1e, all three domains reach their peak performance at CPT fractions between 0.99992 and 0.99994.

Referring to Table 5, our optimal split, capped at 30B total tokens and 1.8M to 2.4 SFT tokens, demonstrates that competitive or even superior performance can be achieved with up to a **95% reduction** in total CPT and SFT tokens. Notably, it **outperforms Code Llama and Code Llama - Python** (Rozière et al., 2023) by **11.6% and 6.7%** on the HumanEval pass@1 datasets, of which 600B training tokens were used for coding adaptation. Moreover, in the health domain, our data allocation **outscored Hippocrates-Meta 7B (De et al., 2024) by 6.4%** and **nearly matched the performance of Hippocrates-Mistral 7B** on MedQA (0 shot), where they used 58.7M tokens for SFT.

### 4.1 ABLATIONS

Corroborating with common approaches of domain adaptation that rely solely on SFT (Luu & Buehler, 2025), results reveal that at the CPT fractions corresponding to the predicted peaks, indicated by (P) in Tables 1, 2, and 3, the performance of our optimal CPT:SFT split consistently surpasses the SFT-only ablation results. Specifically, for Chemistry (CPT fraction 0.99993), Coding (0.999924), and Health (0.9999327), the benchmark scores achieved with the mixed CPT:SFT allocation exceed those obtained when only SFT is applied. This observation reinforces our claim that these allocations represent near-optimal splits for maximising downstream task performance across all three domains.

Table 1: SFT Ablation Results for the Chemistry Domain (ChemBench).

| CPT Fraction | ChemBench Score | SFT Ablation |
|---|---|---|
| 0.99993 (P) | 45.5875 | 45.0615 |
| 0.999934 | 43.4874 | 43.7554 |
| 0.999937 | 44.3723 | 43.6302 |

Table 2: SFT Ablation Results for the Coding Domain (HumanEval and HumanEval+).

| CPT Fraction | HumanEval Score | HumanEval+ Score | HumanEval SFT Ablation | HumanEval+ SFT Ablation |
|---|---|---|---|---|
| 0.999924 (P) | 0.451219 | 0.39634 | 0.402439 | 0.353659 |
| 0.999926 | 0.432927 | 0.384146 | 0.439024 | 0.378048 |
| 0.999928 | 0.432926 | 0.37804 | 0.426829 | 0.378048 |

## 4.2 MATHEMATICAL MODELLING OF OPTIMUM

A core finding of our research is the discovery of a consistent and remarkably narrow optimal operating range for the CPT:SFT data allocation. Across three diverse domains - Coding, Health, and Chemistry - our rigorous, controlled experiments consistently show that peak downstream performance is achieved when the CPT fraction lies between 0.99992 and 0.99994. This suggests a fundamental "sweet spot" for the number of SFT tokens required for effective alignment, which our experiments place between approximately 1.5 and 2.4 million tokens.

To formalise this finding and obtain a more robust estimate of the true optimum, we calibrated a mathematical model to the empirical data. For example, in the Coding domain, the single highest score was observed at a CPT fraction of 0.999924. However, any individual run is subject to stochastic noise. Our model, by considering the entire trend of all data points, calculates a more reliable peak at 0.999923, effectively averaging out experimental noise to find the most probable centre of the optimal range.

## 4.3 CONSISTENCY ACROSS VARYING TOKEN BUDGETS

To verify if optimal CPT fraction would shift predictably for different total token budgets, we conducted out-of-sample validation experiments for 25B and 35B budgets. Using Equation 2, our universal model predicted the new optimal splits to be `0.999907` for the 25B budget and `0.999934` for the 35B budget.

The subsequent empirical results, summarised in Table 4, provide strong validation for the model's predictive power. The validation for the **35B budget was highly successful**: the empirically observed performance peak occurred at a CPT fraction of **0.999914**, representing a deviation of only **0.00002** from the model's prediction. This remarkable accuracy confirms that our model has captured a fundamental principle of the learning dynamics in this regime.

For the 25B budget, the empirically observed peak was at 0.99975. While this shows a larger deviation from the prediction, the performance curve was notably flatter, with multiple points (0.99950, 0.99975, 0.99990) yielding similar high scores. This suggests that at smaller budgets, the system is more sensitive and the optimal operating zone is broader.

Crucially, the model's success in the 35B case validates that our framework is not merely an explanatory fit but a genuine predictive tool, capable of forecasting the optimal data allocation strategy for new resource regimes with a high degree of precision.

## 5 DISCUSSION

### 5.1 MODEL DRIFT ANALYSIS

To understand the mechanisms driving these performance outcomes, we conducted an *inter-ratio drift analysis* focused on the model's internal representational dynamics. The core of this analysis rests on a simple trade-off: as we increase the CPT:SFT ratio, CPT data points are added while SFT data points are removed. CKA has been established as a robust measure of representational similarity Kornblith et al. (2019); we calculate CKA between pairs of fully trained models (post-CPT and post-CPT+SFT) with different ratio configurations to measure how representational structure

Table 3: SFT Ablation Results for the Health Domain (MedQA).

| CPT Fraction | MedQA Score | SFT Ablation |
|---|---|---|
| 0.99993 | 0.54438 | 0.56009 |
| 0.9999327 (P) | 0.57031 | 0.56559 |
| 0.999934 | 0.55381 | 0.56245 |

Table 4: Predicted and tested CPT fractions with corresponding observed scores for different token budgets in the Coding domain. Predicted peaks are marked with (O), and model validation errors are also reported.

| Token Budget | CPT Fraction | Observed Score | Empirical Optimum | Error (CPT Fraction) |
|---|---|---|---|---|
| 25B | 0.99930 | 0.4146 | | |
| 25B | 0.99950 | 0.4329 | | |
| 25B | 0.99975 | 0.4329 | 0.999750 | 0.000157 |
| 25B | 0.99985 | 0.4146 | | |
| 25B | 0.99990 | 0.4268 | | |
| 25B | 0.99992 | 0.3720 | | |
| 35B | 0.99950 | 0.4146 | | |
| 35B | 0.99970 | 0.4268 | | |
| 35B | 0.99990 | 0.4268 | 0.999914 | **0.000020** |
| 35B | 0.999914 | 0.4512 | | |
| 35B | 0.99993 | 0.3963 | | |
| 35B | 0.99995 | 0.4329 | | |

changes with data allocation and visualize how that aligns with model performance. CKA was calculated over four defined data categories: **(1)** **CPT New Added** - foundational knowledge introduced between models, **(2)** **CPT Replay Added**, **(3)** **SFT New Removed** - novel instruction tuning data absent in the successor model, **(4)** **SFT Replay Removed**. By evaluating these fully trained model pairs on the differential data, i.e., the examples that changed between their training configurations, we directly measure how models adapt their representations to different CPT:SFT ratios and how these adaptations correlate with downstream performance. Our analysis measures representational drift, calculated as $1 - \text{CKA}$. This metric provides two key insights: low drift on removed data signifies robust generalisation, whereas high drift on added data confirms that the model is substantively incorporating new knowledge by updating its internal representations. For computational tractability, this analysis used a fixed random sample of 30% of the differential data within each category. The consistency of patterns across domains suggests this sampling was sufficient to capture the core representational dynamics.

Differential CKA analysis reveals striking asymmetries in data processing. While *CPT New Added* generally shows moderate drift across domains, *CPT Replay Added* exhibits dramatic instability at larger ratios—both CS and Chemistry reach *CPT Replay Added* drift values of 0.31 between 0.999937-0.99994 configurations. This divergence rules out simple capacity constraints and instead suggests that excessive replay could impose interference, a finding consistent with research showing improper replay ratios harm performance Lopez-Paz & Ranzato (2017). Performance stability amidst this turmoil suggests compensation via alternative pathways. This underscores the need for asymmetric processing of new and replay data for optimal continual learning Liu et al. (2024b).

The CPT+SFT model testing reveals different adaptation mechanisms. Health (Figure 2a) shows an extremely large singular *CPT New Added* (red line) drift of 0.345 between the 0.99992-0.999925 configurations which immediately stabilises. In CPT model testing, *SFT Replay Removed* (teal line) also spikes to 0.238 drift between 0.999925-0.999928. These suggest a targeted restructuring and alignment of parameters for the medical domain which is followed by peak performance at 0.9999327, indicating an efficient adaptation. In contrast, Chemistry (Figure 2c) and CS (Figure 2b) exhibit moderate, sustained drift patterns. Chemistry's elevated *SFT New Removed* drift (0.094-0.193, orange line) in the CPT+SFT phase indicates sustained adaptation requirements specific to the chemistry domain. Meanwhile, CS optimises earliest (0.999924) and degrades gradually, possibly reflecting challenges with code generation that benefit from specialized training approaches Chen et al. (2021b) that would benefit from larger SFT exposure.

The critical transition between ratios 0.999937-0.99994 marks a fundamental shift in model behaviour for CS and Chemistry domains. Computer Science (Figure 2b) exhibits extreme volatility in this window, with *CPT Replay Added* drift (blue line) reaching 0.312 and *SFT Replay Removed* drift (teal line) spiking to 0.238 in the CPT phase, coinciding with steep performance decline. Chemistry

Table 5: Performance comparison of our optimised data split versus baseline models, highlighting token budget and CPT:SFT allocation.

| Domain | Model | Total Tokens | CPT Tokens | SFT Tokens | Benchmark | Score % |
|---|---|---|---|---|---|---|
| Coding | Our Split | 30B | ≈29.99B | 1.8M-2.4M | HumanEval pass@1 | 45.1 |
| Coding | Code Llama | 600B | N/A | 20B | HumanEval pass@1 | 33.5 |
| Coding | Code Llama - Python | 600B | N/A | 120B | HumanEval pass@1 | 38.4 |
| Health | Our Split | 30B | ≈29.99B | 1.8M-2.4M | MedQA 0-shot | 57.0 |
| Health | Hippocrates-Meta 7B | N/A | N/A | 58.7M | MedQA 0-shot | 50.6 |
| Health | Hippocrates-Mistral 7B | N/A | N/A | 58.7M | MedQA 0-shot | 59.2 |

Table 6: Model performance on MedQA, HumanEval, HumanEval Plus, and ChemBench4K across different CPT fractions.

| CPT Fraction | MedQA % | HumanEval % | HumanEval Plus % | ChemBench4K |
|---|---|---|---|---|
| 0.99992 | 0.5475 | 0.4024 | 0.3719 | 41.0094 |
| 0.999924 | 0.5562 | 0.4512 | 0.3963 | 41.3379 |
| 0.999926 | 0.5467 | 0.4329 | 0.3841 | 43.6625 |
| 0.999928 | 0.5444 | 0.4329 | 0.3780 | 41.7389 |
| 0.99993 | 0.5703 | 0.3902 | 0.3476 | 45.5875 |
| 0.999934 | 0.5538 | 0.4024 | 0.3719 | 43.4874 |
| 0.999937 | 0.5491 | 0.3720 | 0.3232 | 44.3723 |
| 0.99994 | 0.5397 | 0.3659 | 0.3171 | 42.4340 |
| 0.999943 | 0.5546 | 0.3476 | 0.3110 | 41.3879 |
| 0.9999462 | 0.5373 | 0.3780 | 0.3293 | 42.7297 |
| 0.999947 | 0.5467 | 0.3598 | 0.3110 | 39.5783 |
| 0.99995 | 0.5381 | 0.3841 | 0.3293 | 42.1283 |
| 0.99996 | 0.5302 | 0.3476 | 0.3232 | 41.9980 |
| 0.99997 | 0.5405 | 0.3841 | 0.3292 | 41.9181 |
| 0.99998 | 0.5310 | 0.3963 | 0.3476 | 40.6075 |
| 0.99999 | 0.5067 | 0.3110 | 0.2805 | 39.1378 |

(Figure 2c) shows similar *CPT Replay Added* drift instability (0.312, blue line) in the same interval, with multiple metrics converging before diverging, suggesting a reorganisation point.

Health (Figure 2a), however, follows a distinctly different temporal pattern. Its major reorganisation occurs early—between 0.99992-0.999924—and by the 0.999937-0.99994 interval, Health maintains low drift values (0.01-0.09 across metrics); Although interestingly, the CPT model appears to have an reorganisation in this interval as well, albeit at a smaller intensity (possibly suggesting that SFT on top of CPT was able to stabilize and alleviate such a transition). This temporal shift suggests Health successfully completes adaptation before CS and Chemistry encounter their critical instability threshold, avoiding the prolonged volatility observed in other domains.

These domain-specific patterns indicate that while CS and Chemistry reach a critical balance point around 0.999937-0.99994 where the SFT signal might have become insufficient to maintain stable instruction-following pathways, Health's smaller domain gap (see Appendix C) enables earlier, more efficient adaptation.

However, it should be noted that optimal performance consistently aligns with moderate drift (not minimal drift) before and after these critical transitions (Figure 2), validating an optimal drift range found in neural networks Aitken et al. (2022). More saliently, this provides a direct mechanistic explanation for our primary finding and allows us to posit that our range of interest accommodates a composition that balances adaptive flexibility with structural coherence, a concept supported by studies on loss landscapes and implicit regularization Fort & Scherlis (2019); Schoonover & Abbott (2024). These findings reinforce our initial experimental observations from a model similarity perspective, demonstrating that there exists an optimal CPT/SFT allocation that emerges from fundamental mathematical properties of two-phase training Liu & Ueda (2022); Luu & Buehler (2025). Even across domain idiosyncrasies (see Appendix C), our 0.99992–0.99994 range appears to capture

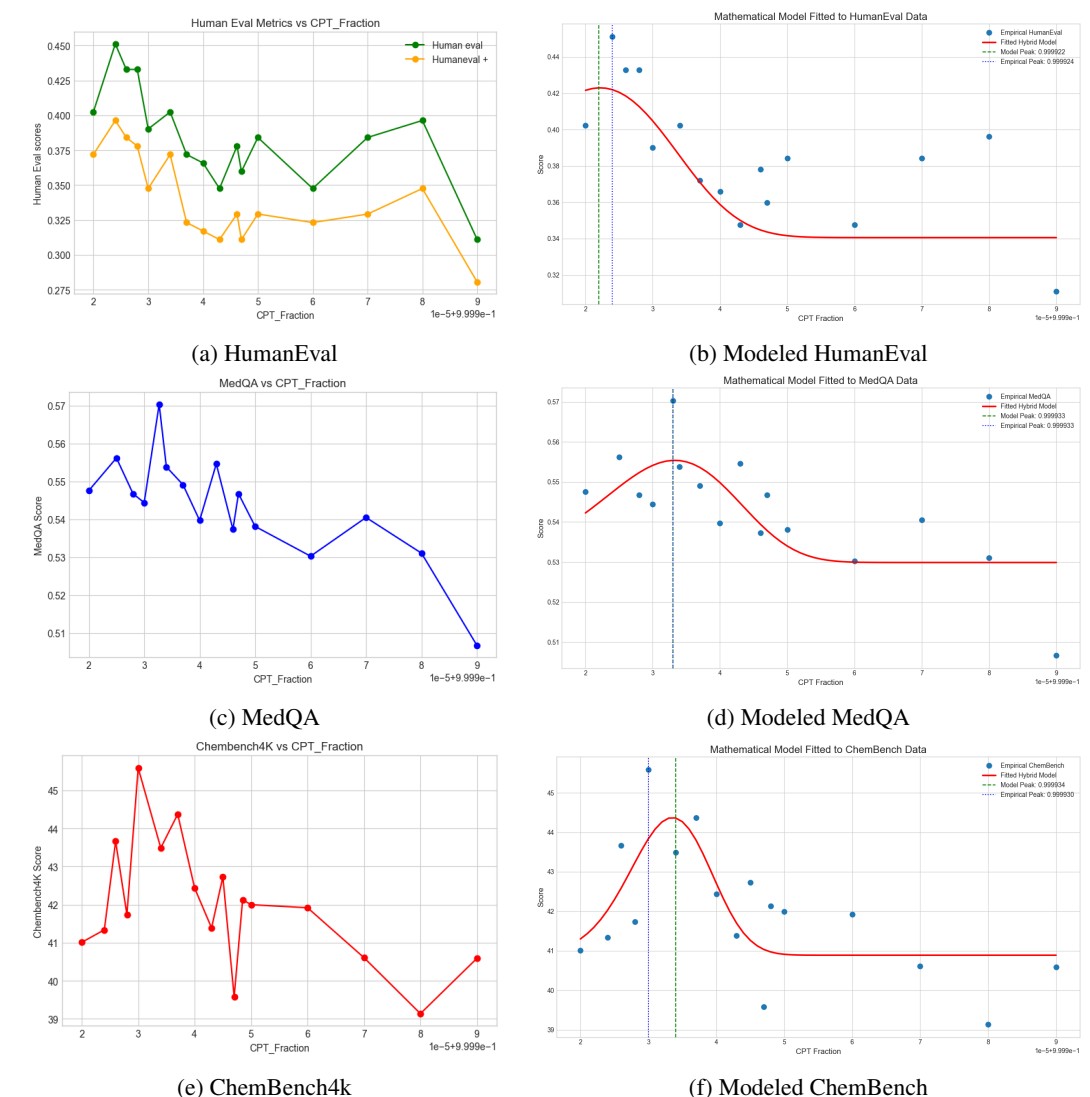

Figure 1: Performance comparison between original and mathematical variants for HumanEval, MedQA, and ChemBench4k.

this region with a general trend of reorganisation, relative stability, and then interference. Perhaps more broadly, this suggests that effective domain adaptation requires a fine balance between representational flexibility for new knowledge and structural coherence for instruction follow-up.

# 6 FUTURE WORK AND LIMITATIONS

Our experiments were limited to 7B-parameter models with 25B-40B token budgets, leaving key scaling questions unanswered. Future work should investigate whether the optimal CPT:SFT ratio shifts with model size—smaller models may need proportionally more SFT data, while larger models could require even less than our observed 1.8-2.4M tokens. Most critically, we must determine if the 0.999937 stability threshold represents a fundamental property of two-phase training that persists at massive scales (100B-1T tokens), or if extreme scales introduce new dynamics requiring proportionally larger SFT datasets to align vast CPT knowledge. Additionally, dynamic scheduling strategies that gradually transition from CPT to SFT could potentially avoid the sharp performance cliffs we observed at fixed ratios, while understanding data quality effects would improve real-world applicability. Developing a complete mathematical theory for why the 0.99992-0.99994 range emerges

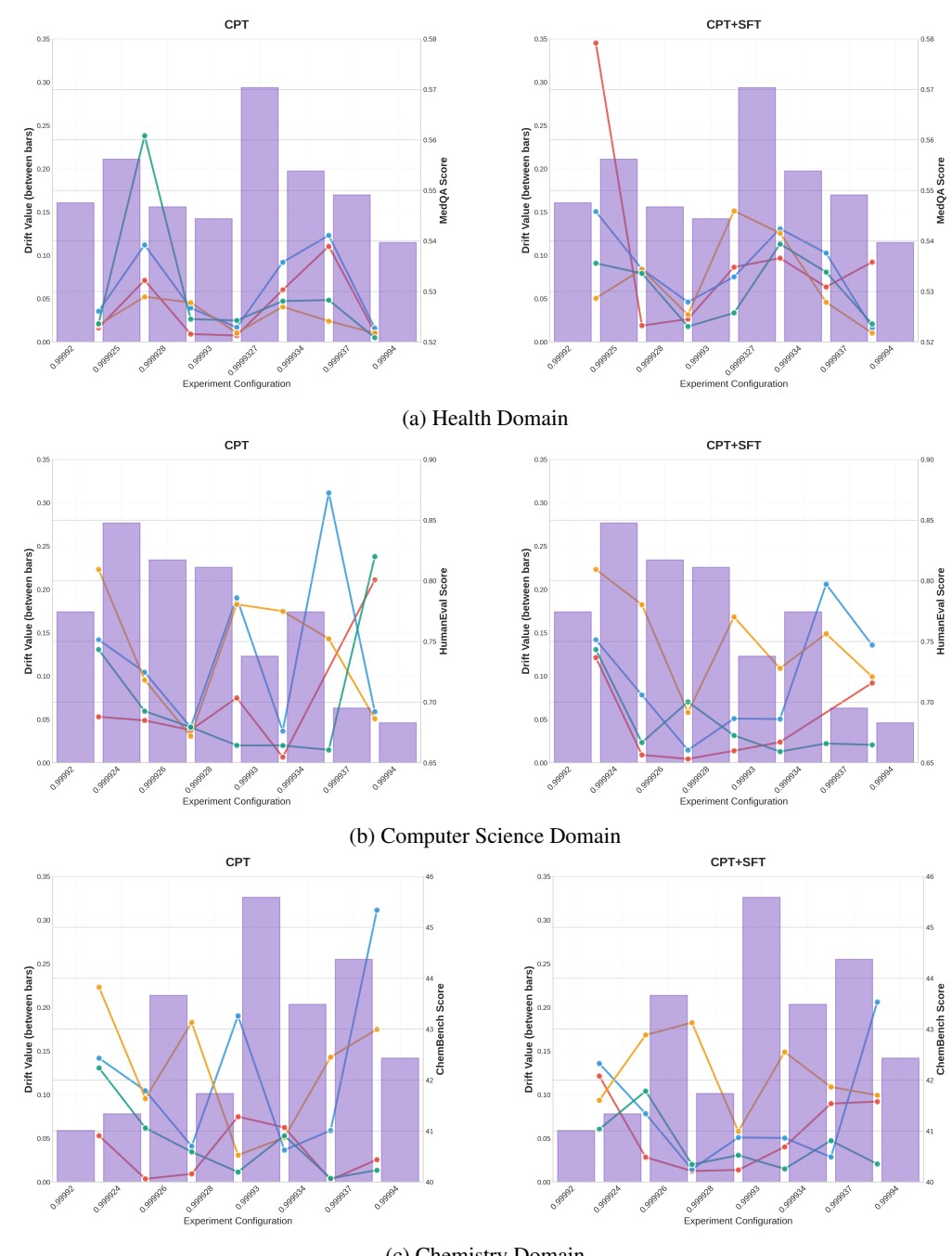

(a) Health Domain

(b) Computer Science Domain

(c) Chemistry Domain

Figure 2: **Inter-Ratio Drift Analysis on SFT Data with CPT only and CPT+SFT trained models.** Performance scores shown as purple bars (right y-axis): **(a)** MedQA, **(b)** HumanEval, **(c)** ChemBench. Colored lines (left y-axis) indicate CKA drift metrics: ● CPT New Added, ● CPT Replay Added, ● SFT New Removed, ● SFT Replay Removed. Peak performance occurs at: (a) 0.9999327, (b) 0.999924, (c) 0.99993. CS and Chemistry exhibit maximum instability between 0.999937-0.99994, while Health shows early reorganisation between 0.99992-0.999925.

universally would transform our empirical findings into predictive principles, enabling analytical determination of optimal allocations without extensive experimentation.

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

## REPRODUCIBILITY STATEMENT

To ensure the reproducibility of our findings, we provide a comprehensive overview of our experimental assets. All experiments were based on the publicly available Mistral-7B-v0.3 foundation model. The training pipeline was implemented using the open-source mistral-finetune library. Our full experimental code, including data processing scripts, the mathematical modeling framework, and the shell scripts used to launch experiments, will be made publicly available upon publication.

The datasets used for both Continual Pre-training (NVIDIA ClimbLab) and Supervised Fine-tuning (BigBio MedQA, Open Code Instruct, ChemData700k) are all publicly accessible. For all benchmark evaluations, we strictly adhered to the official train/test splits to prevent data leakage. A complete list of all hyperparameters, including optimiser settings and LoRA configurations for both the CPT and SFT stages, is detailed in Section 3 and the Appendix. All experiments were conducted on NVIDIA H100 GPUs.

## A    APPENDIX

Table A.1: CPT Ablation with Larger Step Sizes and ChemBench4k.

| CPT Steps | ChemBench4k |
|---|---|
| 4000 | 41.5461 |
| 6000 | 39.1529 |
| 8000 | 42.0371 |
| 10000 | 39.6895 |

Table A.2: Continual Pre-Training (CPT) Stage: Dataset and Hyperparameters.

| Aspect | CPT Stage |
|---|---|
| Dataset | NVIDIA ClimbLab Clusters 7, 11, 17 (subset ¡ 30B tokens) |
| Sequence Length | 16,384 |
| Global Batch Size | 3 |
| Per-Device Batch Size | 3 (H100 GPU) |
| Total Training Steps | 2,000 |
| Optimiser | AdamW |
| Learning Rate | $6 \times 10^{-5}$ (constant) |
| Learning Rate Schedule | Linear warmup (first 5% of steps) |
| Weight Decay | 0.1 |
| LoRA Rank | 256 |
| LoRA Scaling Factor ($\alpha$) | 512 |
| LoRA Dropout | 0.0 |
| Gradient Clipping | Max norm 1.0 |
| Random Seed | Fixed for reproducibility |

Table A.3: Supervised Fine-Tuning (SFT) Stage: Dataset and Hyperparameters.

| Aspect | SFT Stage |
|---|---|
| Dataset | BigBio MedQA (Health), Open Code Instruct (Python Coding), ChemData700k (Chemistry) |
| Sequence Length | 16,384 |
| Global Batch Size | 3 |
| Per-Device Batch Size | 3 (H100 GPU) |
| Total Training Steps | 300 |
| Optimiser | AdamW |
| Learning Rate | $6 \times 10^{-5}$ (constant) |
| Learning Rate Schedule | Linear warmup (first 5% of steps) |
| Weight Decay | 0.1 |
| LoRA Rank | 64 |
| LoRA Scaling Factor ($\alpha$) | 128 |
| LoRA Dropout | 0.0 |
| Gradient Clipping | Max norm 1.0 |
| Random Seed | Fixed for reproducibility |

Table A.4: Overview of NVIDIA ClimbLab Dataset Clusters Used for Continual Pre-training (CPT).

| Cluster ID | Tokens (Billions) | Primary Topics |
|---|---|---|
| 7 | 64.04 | Chemistry, Insects, Taxonomy, Agriculture, Veterinary Science |
| 11 | 37.11 | Software Development, Programming, Web Development, JavaScript |
| 17 | 82.23 | Cardiovascular Health, Medical Research, Immunology, Cancer |

Table A.5: Overview of Supervised Fine-Tuning (SFT) Datasets Used.

| Dataset | Tokens | Domain |
|---|---|---|
| BigBio MedQA | 2,742,003 | Health |
| Open Code Instruct | 2,124,508,238 | Python Coding |
| ChemData700k | 146,577,757 | Chemistry |

## B  BENCHMARK DATASETS

**HumanEval** (Chen et al., 2021a) is a code generation benchmark that assesses the functional correctness of Python programs generated by LLMs. It consists of a series of programming problems with associated unit tests, requiring models to produce code that satisfies the specifications. The benchmark is designed to evaluate both the syntactic correctness of the generated code and its ability to solve practical programming tasks.

**HumanEval Plus** (Liu et al., 2023) extends the original HumanEval dataset by introducing more complex programming problems, additional problem types, and edge-case scenarios. This benchmark aims to provide a more rigorous evaluation of LLMs' code generation capabilities, testing not only correctness but also reasoning, comprehension, and problem-solving skills in Python coding tasks.

**ChemBench4k** (Zhang et al., 2024) is a chemistry-specific benchmark designed to evaluate models on tasks requiring domain knowledge in chemical structures, molecular properties, and chemical reactions. The dataset consists of 4,000 curated tasks that assess a model's ability to predict molecular attributes, understand chemical relationships, and apply chemical reasoning. ChemBench4k is particularly useful for testing specialised knowledge in computational chemistry and drug discovery.

**MedQA** (Jin et al., 2021) presents questions in the format of the US Medical Licensing Examination (USMLE) and covers a wide range of medical knowledge, including patient profiles, disease symptoms, treatment protocols, and drug dosage calculations. The training set contains 10,178 questions, while the test set includes 1,273 questions. MedQA is offered in two configurations: a four-choice format (MedQA) and a five-choice format (MedQA 5-options). In our evaluation, the former was used.

**MMLU** (Hendrycks et al., 2021) is a general-purpose benchmark comprising exam questions spanning 57 subjects, including philosophy, management, STEM, and medical domains. For our evaluation, we mainly assess generalisation performance in all tasks.

**HellaSwag** (Zellers et al., 2019) evaluates commonsense reasoning in natural language understanding. The benchmark consists of multiple-choice questions where each item presents a context and several candidate continuations, requiring the model to select the most plausible continuation. HellaSwag is particularly challenging because many options are syntactically correct, and success requires understanding of real-world knowledge, cause-and-effect reasoning, and narrative coherence.

## C    INTER-PHASE MODEL DRIFT ANALYSIS

We conducted inter-phase drift analysis using BASE→CPT and CPT→SFT model comparisons on a fixed random sample of 200 examples from each domain's SFT dataset across all experimental configurations. This analysis reveals how domain characteristics implicitly determine optimal CPT/SFT ratios.

BASE→CPT drift magnitude could be used as a proxy for the domain gap (loosely: the difference in alignment between a base model and a specialized domain) which would fundamentally shape training dynamics. Chemistry exhibits the largest gap (30-37% BASE→CPT drift), requiring aggressive CPT allocation—performance peaks at 0.99993 coincide with relatively stable BASE→CPT drift regions, as the model must preserve hard-won domain knowledge. Health shows a smaller domain gap (12-16% drift), allowing CPT→SFT dynamics more influence, with performance peaking at 0.99993 despite drift variations. Computer Science demonstrates the clearest pattern: performance peaks at 0.999924 when BASE→CPT drift is minimal (∼16%), then degrades sharply as drift increases to 28% at higher ratios, showing strong inverse correlation.

These domain characteristics explain systematic variation in optimal ratios. Large, persistent domain gaps necessitate higher CPT fractions to maintain stability (Chemistry at 0.99993 with 30%+ drift throughout). Domains with variable gaps benefit from finding the sweet spot of minimal drift (CS at 0.999924). Health's consistently moderate gap enables stable performance across wider ratio ranges. These patterns suggest that the magnitude of the domain gap and volatility are key factors that implicitly skew optimal ratios, though dataset-specific characteristics may also contribute.

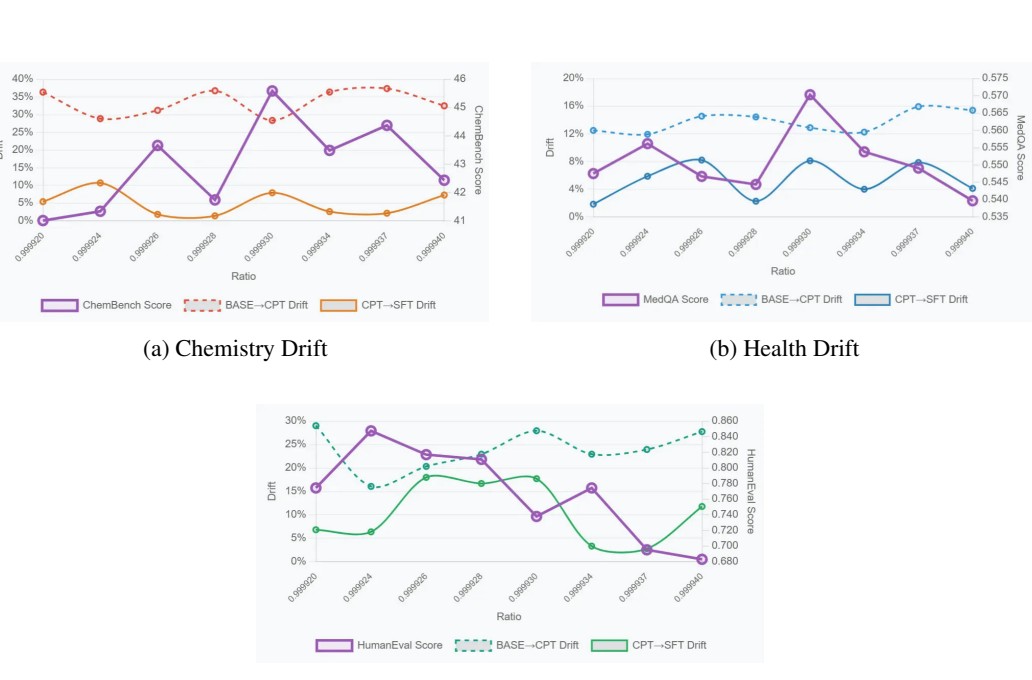

(a) Chemistry Drift

(b) Health Drift

(c) Computer Science Drift

Figure 3: **Inter-Phase Model Drift Analysis Across CPT/SFT Ratios.** BASE→CPT drift (dashed lines) quantifies domain gap; CPT→SFT drift (solid lines) shows adaptation dynamics. Performance in purple. **(a) Chemistry**: Large domain gap ($>30\%$) with performance peaks at minimal BASE→CPT drift. **(b) Health**: Small domain gap ($\sim12\%$) with performance driven by CPT→SFT dynamics. **(c) Computer Science**: Moderate gap with inverse correlation between BASE→CPT drift and performance.

