# OpenReview forum: "Modelling Optimal Trade-Off Between Continued Pre-Training and Supervised Fine-Tuning for LLM Domain Adaptation"
_ICLR.cc/2026/Conference — ICLR 2026 Conference Withdrawn Submission_

### Official Review · Reviewer_Yftf · 2025-10-30

**Soundness:** 1
**Presentation:** 2
**Contribution:** 1
**Rating:** 0
**Confidence:** 5

**Summary:**

The paper explores how to split a fixed token budget between continual pre-training (CPT) and supervised fine-tuning (SFT) when adapting large language models to specific domains. Using Mistral-7B across medicine, chemistry, and coding, the authors test different CPT:SFT ratios and claim performance peaks when almost all tokens go to CPT and only about 2 million to SFT. They fit these results with a custom mathematical model combining a Gaussian “performance peak” and a reciprocal “failure cliff,” proposing a universal optimal ratio (~0.99993 CPT fraction) and presenting it as a general principle for efficient domain adaptation.

**Strengths:**

The paper tackles an underexplored question, how to optimally divide training tokens between continual pre-training and supervised fine-tuning, offering a fresh perspective on data allocation in LLM domain adaptation.

If validated, the idea of a predictable CPT:SFT trade-off could help guide more efficient use of compute and data for domain-specific LLM adaptation.

**Weaknesses:**

The reported “optimal ratio” (CPT fraction ≈ 0.99993) is a direct consequence of the artificially small SFT cap rather than an emergent property of the model or data. The finding reflects experimental design, not a general principle of LLM training.

The proposed Gaussian + reciprocal model is purely post-hoc curve fitting with six free parameters and no theoretical justification. Presenting it as a “predictive law” is misleading.

Results rely on single runs per ratio with noisy scores, lacking confidence intervals or multiple seeds. The observed performance differences are within expected training variance.

Comparisons to fully trained models such as Code Llama or Hippocrates-Mistral use different architectures and token scales, invalidating claims of “95 % budget reduction” or superior performance.

**Questions:**

1. How were the reported 25–35 B token budgets calculated, given that the listed training steps, sequence lengths, and batch sizes correspond to roughly 100 M tokens in total?

2. How do the authors justify comparing LoRA-based adaptations, where only a small subset of parameters is updated, to fully retrained models such as Code Llama or Hippocrates-Mistral?

3. Why was the SFT dataset limited to approximately 2 M tokens, was this a deliberate design choice or a data constraint, and if deliberate, doesn’t this effectively predetermine the observed CPT fraction (~0.99993) rather than reveal it empirically?

4. What theoretical justification supports the use of a log-Gaussian combined with a reciprocal term to model performance as a function of SFT token count?

5. On what basis do the authors describe the fitted Gaussian center as a fixed “property of the adaptation task,” and what evidence suggests this parameter generalizes beyond the specific experiments reported?

6. Will the authors provide the full code, training logs, and checkpoints necessary to independently verify token counts and evaluation results?

7. Given the inconsistencies in token accounting and the absence of repeated trials, what grounds do the authors have for asserting the existence of a “universal optimum” at a CPT fraction of 0.99992–0.99994?

---

### Official Review · Reviewer_d11b · 2025-10-31

**Soundness:** 2
**Presentation:** 3
**Contribution:** 2
**Rating:** 4
**Confidence:** 4

**Summary:**

The paper studies how to allocate a fixed training-token budget between Continual Pre-Training (CPT) and Supervised Fine-Tuning (SFT) for domain adaptation. Using Mistral-7B and a 30B-token budget, the authors sweep 16 CPT:SFT allocations across health, chemistry, and coding, finding a tight optimum at CPT fraction ≈ 0.99992–0.99994 (about 1.8–2.4M SFT tokens) for all three domains. They propose a hybrid mathematical model (Gaussian peak + reciprocal “alignment cliff”) that fits the ratio–performance curve and predicts optima for different total budgets (validated at 25B and 35B). They further analyze representation changes via CKA-based drift, arguing peak performance coincides with “moderate drift” rather than minimal drift.

**Strengths:**

(1)  Clear, new problem framing. The paper directly targets the neglected CPT:SFT allocation question under a fixed budget, moving practice beyond ad-hoc splits.
(2)  Consistent empirical finding across domains. The same narrow optimum (~0.99993 CPT fraction) appears for coding/health/chemistry, suggesting a robust phenomenon rather than a single-domain quirk.

**Weaknesses:**

(1)  Limited external validity (scale & regime). All experiments use Mistral-7B and ≤ 35B tokens. It remains unclear whether the same optimum and the reported “stability threshold” persist for larger models (e.g., 70B–180B) or 100B–1T-token adaptation, or other model families, which the authors acknowledge.
(2)  The Gaussian + reciprocal form fits well and generalizes across budgets, but it is calibrated to data rather than derived from learning theory; the paper itself calls for a fuller mathematical account in future work.
(3)  The SFT datasets are intentionally tiny (≈1.8–2.4M tokens); CPT uses PEFT/LoRA rather than full-parameter training. Benchmarks emphasize domain tasks (HumanEval/+, MedQA, ChemBench4k). These reasonable choices for cost-efficiency make it uncertain whether the ~0.99993 optimum holds with richer SFT, broader task mixes, or full-parameter CPT; stronger multi-stage alignment baselines could also shift conclusions.

**Questions:**

Since your optimal ratio holds under a fixed 2 M-token SFT cap, how does data *quality* versus *quantity* affect the curve? Would higher-quality instruction data flatten or shift the optimum?

---

### Official Review · Reviewer_tZgs · 2025-11-02

**Soundness:** 2
**Presentation:** 2
**Contribution:** 2
**Rating:** 2
**Confidence:** 4

**Summary:**

This paper investigates the optimal allocation of training tokens between Continual Pre-training (CPT) and Supervised Fine-Tuning (SFT) for domain adaptation of Large Language Models under a fixed 30B token budget. The authors conduct experiments across three domains (health, chemistry, and coding) using 7B-parameter models, testing 16 different CPT:SFT ratios.
Key findings show that optimal performance consistently occurs at CPT fractions between 0.99992-0.99994 across all domains, corresponding to approximately 29.998B CPT tokens and 1.8-2.4M SFT tokens. The authors develop a mathematical framework combining Gaussian and reciprocal functions to model this relationship and validate it across different token budgets (25B-35B). The work provides practical guidance for efficient domain adaptation by demonstrating that minimal SFT data (under 0.01% of total budget) is sufficient for effective alignment.

**Strengths:**

1. This is the first systematic study examining the trade-off between CPT and SFT under fixed token budgets, addressing a critical gap in domain adaptation research. The work tackles a genuine practical problem that practitioners face when allocating limited computational resources between knowledge acquisition (CPT) and task alignment (SFT).
2. The authors conduct rigorous experiments across three diverse domains (health, chemistry, coding) with 16 different CPT:SFT ratios, demonstrating consistency in their findings. The inclusion of validation experiments at different token budgets (25B, 35B) and comparison with established baselines provides solid empirical evidence for their core claims

**Weaknesses:**

1. The authors provide only intuitive explanations for Equation 1's structure without theoretical grounding. No justification is given for why performance should follow a Gaussian combined additively with a reciprocal cliff, nor why alternative functional forms such as power law, exponential decay were rejected.
2. The claimed optimal CPT fractions of 0.99992-0.99994 (spanning only 2×10⁻⁵) require unrealistic precision without proper statistical validation. The study lacks confidence intervals, multiple runs, or robustness analysis to support such sensitive claims that would be impractical in real-world scenarios.
3. All experiments are restricted to 7B parameter models, severely limiting applicability to larger, industry-relevant models (70B+).  There is no evidence that the "universal" optimal ratios hold across different model scales, which makes generalizability difficult. The authors do acknowledge this limitation.

**Questions:**

1. Parameters like A, \mu and \sigma are srbitrary in the mathematical formulation of equation 1. Since there are limited data points for fitting to an equation the form of the equation needs good theoretical backing to be used as the ansatz. With fewer points any function form can be fitted. What is the rationale behind the chosen form of equation 1 ?

2. The CPT fractions between 0.99992-0.99994 is reported without any confidence intervals or error bars. The repeatability of the experiments is missing in the absence of multiple runs.

3. To complete the empirical study models in the parameter range of at least 70B should be experimented with. Its likely that the quoted "universal" ratio will break at larger scales though a functional estimate should be possible which would make the work complete.

---

### Note · Authors · 2025-11-28

I have read and agree with the venue's withdrawal policy on behalf of myself and my co-authors.